# Modern Bronchoscopic Treatment Options for Patients with Chronic Bronchitis

**DOI:** 10.3390/jcm12051854

**Published:** 2023-02-26

**Authors:** Anna Katharina Mayr, Arschang Valipour

**Affiliations:** 1Karl Landsteiner Institute for Lung Research and Pulmonary Oncology, 1210 Vienna, Austria; 2Department of Respiratory and Critical Care Medicine, Klinik Floridsdorf, 1210 Vienna, Austria

**Keywords:** Chronic Obstructive Pulmonary Disease (COPD), chronic bronchitis, bronchial rheoplasty, metered cryospray, balloon desobstruction, targeted lung denervation

## Abstract

Chronic Obstructive Pulmonary Disease (COPD) is one of the leading causes of death worldwide and has a large impact on a patient’s quality of life due to its wide range of symptoms and comorbidities. There are known to be different phenotypes in COPD with various extents on the burden of the disease and its prognosis. Chronic bronchitis with persistent cough and mucus production is regarded as one of the main symptoms of COPD with tremendous effects on subjectively reported symptom burden and frequency of exacerbations. Exacerbations in turn are known to have an impact on disease progression and increase health care costs. Modern bronchoscopic treatment options are currently under investigation targeting the problem of chronic bronchitis and frequent exacerbations. This review summarizes the existing literature about these modern interventional treatment options and provides perspectives on upcoming studies.

## 1. Introduction

According to the World Health Organization (WHO) Global Health Estimates, Chronic Obstructive Pulmonary Disease (COPD) was the third leading cause of death worldwide in 2019 [1]. Moreover, COPD is associated with a large burden of disease due to its symptoms and reduced physical activity as well as with an increase in disability-adjusted life years (DALYs) and furthermore increases the financial burden on healthcare providers [1,2].

Chronic bronchitis is defined as chronic cough and sputum production for at least three months per year for two consecutive years [3]. An increased number of goblet and inflammatory cells in the epithelium of peripheral airways is described as the underlying pathophysiologic mechanism, even in smokers without significant airflow obstruction. Furthermore, current smoking status was recognized to be a predictor of goblet cell density in a multivariate analysis [4,5]. The parasympathetic nervous system contributes to this pathophysiology through its main neurotransmitter acetylcholine, which induces bronchoconstriction and mucus secretion and might regulate inflammatory processes [6].

In 2012, the Spanish COPD Guidelines (GesEPOC) proposed one of the first attempts of a phenotype-based therapeutic approach, with a distinction into four different phenotypes of COPD on the basis of the extent of concomitant chronic bronchitis, exacerbation frequency or the presence of concomitant asthma [7]. Since then, the heterogeneity in the disease, the phenotype-specific different impacts on prognosis and individualized therapeutic approaches have been further emphasized [8,9]. In an observational study of 831 COPD patients in Spain, 66.2% of the included population were categorized as non-exacerbators, 15.0% had an overlap with asthma, 4.6% had frequent exacerbations with emphysema and 11.9% with chronic bronchitis; 2.3% were exacerbators but did not fulfill the criteria for any of the groups [10]. Another multinational study of over 3000 stable COPD patients conducted in central and eastern Europe found prevalence rates of frequent exacerbators with or without symptoms of chronic bronchitis at 20.4% and 9.5%, respectively [11]. Importantly, these and other reports found the phenotype with chronic bronchitis to be the most symptomatic with worse quality of life as well as a higher risk of exacerbations and elevated mortality risk [12,13]. As respiratory symptoms such as chronic bronchitis increase the risk for exacerbations even in current or former smokers with preserved lung function, this further emphasizes a phenotype-based therapeutic approach [14].

Exacerbations, in particular, have a negative effect on the patient’s prognosis, with increasing mortality risk associated with the frequency and severity of the events [15]. A more rapid and additional lung function decline has been described in patients with each exacerbation event, even in patients with mild disease [16]. Moreover, exacerbations have a negative effect on the quality of life and contribute to overall higher health care costs [2,17,18]. 

Positive effects of drug and inhaler therapy on exacerbation frequency and rate of hospital admissions have already been described [19,20,21]. This review summarizes the cumulating literature regarding modern bronchoscopic treatment options for patients with COPD and chronic bronchitis with or without frequent exacerbations for a more phenotype-based, individualized therapeutic approach.

## 2. Bronchoscopic Treatment Options for Chronic Bronchitis and Frequent Exacerbations

There are currently three different interventional methods for patients with COPD specifically aimed at targeting chronic bronchitis—the RheOx bronchial rheoplasty (GALA Therapeutics, San Carlos, CA, USA), the RejuvenAir System metered cryospray (CSA Medical, Lexington, MA, USA) as well as the Karakoca resector balloon desobstruction. Furthermore, another method—targeted lung denervation (Nuvaira Inc., Maple Grove, MN, USA)—mainly aims to help frequent exacerbators with or without symptoms of chronic bronchitis.

### 2.1. RheOx Bronchial Rheoplasty

During bronchial rheoplasty nonthermal, high-frequency pulsed electrical energy is delivered to the airway tissue in order to ablate mucus-producing cells (mainly goblet cells) and initiate epithelial regeneration. The intervention is performed under general anesthesia and takes place in two separate steps (one lung per step) with one month between, whereby all of the accessible airways from the subsegmental region to the main carina are treated by stepwise expanding and collapsing the electrode (Figure 1) for circumferential contact to the mucosa [22]. 

The first feasibility and safety data were obtained from two pooled multicenter, single-arm clinical studies enrolling 30 patients with chronic bronchitis [22]. The intervention was conducted as described with an additional histologic evaluation of mucosal cryobiopsies, and patients were followed-up for 12 months, with safety through 6 months defined as the primary outcome. Symptom score and histological effects on goblet cell hyperplasia were further outcomes. Rheoplasty was completed in all patients, and no device-related serious adverse event was noticed. Four procedure-related serious adverse events were reported through 6 months (one pneumonia, one mucosal scarring, two exacerbations) and none afterward. Mild hemoptysis in 47% of the patients was the most frequent non-serious adverse event reported. The goblet cell hyperplasia score was significantly reduced after treatment (*p* < 0.001) and significant improvements in the COPD Assessment Test (CAT; median −8.0; *p* = 0.0002) and St. George’s Respiratory Questionnaire (SGRQ; median −7.2; *p* = 0.0002) were observed after 6 months, with similar treatment effects through further follow-up (Table 1).

Preliminary data of a still ongoing registry study, presented at the 2022 European Respiratory Society (ERS) International Congress, also provided favorable safety results and clinically meaningful improvements in symptoms over follow-up (Table 1) [23].

According to clinicaltrials.gov, there is another active open-label, single-arm study in the U.S. (NCT03631472), one completed in Canada (NCT03385616) and one not yet recruiting in China (NCT05641207). Moreover, a randomized, parallel-group, double-blind, sham-controlled, multicenter trial has already started (NCT04677465).

Overall, bronchial rheoplasty seems to be a promising, feasible intervention for patients with chronic bronchitis regarding the reduction in symptom burden and improvement in quality of life with a good safety profile during a one-year follow-up. Data from the upcoming randomized, sham-controlled trial will provide more information about the effect’s durability and long-term outcomes as well as safety.

### 2.2. RejuvenAir System Metered Cryospray

The RejuvenAir System metered cryospray uses programmed doses of liquid nitrogen, applied through a catheter with a radial spray head inserted through the working channel of the bronchoscope (Figure 2), to circularly cryoablate the abnormal mucosa and goblet cells to a depth of 0.1–0.5 mm and a width of up to 10 mm [24]. Evidence of histologic effects and safety of the procedure come from non-chronic bronchitis patients [25,26]. 

Subsequently, an open-label, single-arm study confirmed the safety, feasibility and efficacy of the intervention specifically in patients with COPD and chronic bronchitis [24]. In this study, the whole treatment took place in three separate steps, each 4–6 weeks apart, and treating one after another: 1. the right lower lobe and main stem bronchus, 2. the left lower lobe and main stem bronchus and 3. both upper lobes, any residual main stem bronchus and the distal end of the trachea. A total of 35 patients were enrolled, and 34 completed all three treatment steps. Overall, 14 serious adverse events were reported up to 12 months, and none were device- or procedure-related. In addition, 6 non-serious device-related (1 bronchospasm during treatment, 5 exacerbations) and 40 non-serious procedure-related adverse events occurred. After 3 months, the SGRQ (−6.4, 95% Confidence Interval CI −11.4 to −1.3; *p* = 0.01), CAT (−3.8, 95% CI −6.4 to −1.3; *p* < 0.01) and Leicester Cough Questionnaire (LCQ; 21.6, 95% CI 7.3 to 35.9; *p* < 0.01) improved significantly and were persistent up to 6 or 9 months. After 12 months, improvements exceeded the minimal clinically important difference (MCID) but did not reach statistical significance (Table 1).

According to clinicaltrials.gov, there is a randomized, cross-over, sham-controlled trial in the state of recruitment (NCT03893370) and another one active but not recruiting (NCT03892694).

In summary, metered cryospray seems to be feasible with an acceptable safety profile over a one-year follow-up and efficacy regarding reduction in symptoms and improvement in quality of life up to nine months. More long-term efficacy data are needed to evaluate the durability of its effects, as no significant improvements in symptom scores were noticed at 12 months. Nevertheless, this might have been due to a long time gap between the first and second treatments, as stated by the authors.

### 2.3. Karakoca Resector Balloon Desobstruction

The Karakoca resector balloon desobstruction uses a latex balloon on a polyethylene tube, covered with a mesh structure, for repeated in- and deflation in the segmental and subsegmental bronchi using an electronic pump, with the intention of mechanically disturbing the altered mucosal layer [28].

A small pilot study in 10 patients with COPD and predominant chronic bronchitis phenotype postulated the feasibility of the intervention [28]. According to the author, no intra- or postoperative complications or exacerbations occurred within a 1- to 3-month follow-up. The report further indicates changes in oxygenation, lung function and the Borg Dyspnea Scale following the procedure, as well as a decline in the number of goblet cells. However, only individual data are presented, and no statistical analysis was performed.

Subsequently, a case series of 188 patients was published by the same authors showing significant (*p* < 0.001) improvements in lung function, the Borg Scale, oxygen saturation and the 6 min walking test (6MWT) in the postinterventional follow-ups at 1 week and 1 month after the procedure (Table 1) [29]. According to the authors, no intra-, peri- or postoperative complications and no exacerbations were seen within 3 months of follow-up. The study comprised a broad range of COPD patients with chronic bronchitis and frequent exacerbations, from outpatient treated patients with home non-invasive mechanical ventilation to ones receiving intensive care treatment due to acute exacerbation. 

In summary, information from longer-term prospective trials is lacking, which does not permit a comprehensive evaluation of this method. According to clinicaltrials.gov, no trials are currently ongoing or in planning.

### 2.4. Targeted Lung Denervation

Targeted lung denervation (TLD) addresses symptomatic patients with frequent exacerbations. A dual-cooled radiofrequency catheter (Figure 3) is used in both main bronchi to circumferentially ablate the parasympathetic vagal fibers innervating the lung, in order to reduce their bronchoconstrictive, mucus-producing and inflammatory effect on the airways [30]. Extensive animal and human cadaver testing were undertaken before in-human testing was commenced [30]. In addition, more efficacy trials were performed in animal models with sheep and dogs [31,32].

The first in-human pilot study observed the safety, feasibility and efficacy in 22 COPD patients using a two-step procedure with two different energy levels (20 W vs. 15 W) over a 1-year follow-up period [30]. Technical feasibility was achieved in 93%, and the primary safety endpoint (freedom from device- or procedure-related, documented and sustained worsening of COPD up to 1 year postinterventional) was met in 95% of patients. Nevertheless, three procedural adjustments were made during the study, due to device-related asymptomatic airway wall effects observed (superficial tissue defects, small perforation of the main carina, granuloma with 20% stenosis; all resolved) in 3 of 12 patients in the 20 W group after energy delivery to the thin, thermally sensitive tissue of the proximal main carina and imperfect cooling-balloon contact to the bronchial walls, resulting in a higher local energy level and thus risk of side effects. Gastroparesis occurred in one individual as a device-related serious adverse event. In summary, a tendency towards better effects on pulmonary function, exercise capacity and quality of life was observed in the higher energy level group (Table 1). In a sub-study investigation, attenuation of airway inflammation could be observed [33] and additional bronchodilatory effects of antimuscarinics to TLD were noted in a post hoc analysis [34].

In the next step, safety and feasibility were evaluated in a prospective study performing TLD during a single bilateral procedure in 15 patients with moderate-to-severe COPD [35]. The primary safety endpoint (freedom from device-related worsening of COPD up to 1-year postintervention) was met in 100% of patients. No significant wall effects were seen during a follow-up bronchoscopy, and the one-step procedure was shown to be feasible. Similar effects on lung function and exercise capacity were demonstrated as in the earlier study [30], and none of the adverse events that occurred for up to 3 years were considered treatment-related (Table 1).

A second-generation device system was then assessed for the first time in 30 individuals with COPD for safety, feasibility and dosing (29 vs. 32 W) in a randomized, double-blind trial (AIRFLOW-1 study) [36]. Four TLD-associated adverse airway effects and five serious gastric events, most likely due to injury of the esophageal vagal branches, occurred in the early randomized phase of the study, after which the trial was stopped, and procedural enhancements were made. These included optimized catheter placement (with better balloon contact) and visualization of the esophagus using an inserted balloon catheter, with the intention of positioning the treatment catheter beyond a prespecified distance from the esophagus and avoiding the medial wall of the right main stem bronchus. An additional 16 patients were enrolled in an open-label study for safety confirmation of the procedural enhancements, which reduced both gastrointestinal and airway events in the subsequent phase of the randomized trial and during the follow-up confirmation study. Furthermore, improvements in lung function and quality of life were observed with a trend towards more favorable outcomes with the higher energy dose (Table 1). Overall, the safety of TLD was considered to be acceptable. In a long-term follow-up over 3 years, no treatment-related late-onset serious adverse events occurred and lung function and quality of life, as well as the rate of exacerbations, remained stable [37]. A post hoc analysis including preinterventional lung function measurements showed a 66% lower decline in forced expiratory volume in 1 s (FEV_1_) after treatment compared to the preinterventional course [38]. 

Within the comprehensive clinical trial program, a multicenter, randomized, sham-bronchoscopy-controlled, double-blind trial (AIRFLOW-2) with 82 symptomatic COPD patients was conducted [39]. The findings from this report demonstrate a significant reduction in the rate of respiratory events (including exacerbation rate) between the 3- and 6.5-months follow-up visit in the TLD group compared to the sham group (32% vs. 71%, *p* = 0.008). Moreover, a significant reduction in the risk of exacerbation requiring hospitalization up to 12.5 months was observed (hazard ratio 0.35, 95% CI 0.13–0.99; *p* = 0.039) (for other clinical results see Table 1). The time to first severe exacerbation was significantly lengthened in the treatment group (hazard ratio 0.38, *p* = 0.04) over a 2-year follow-up period (for other clinical results see Table 1) [40]. Similar results, although not significant, were seen after unblinding, cross-over and treatment of subjects in the initial sham group, which possibly resulted from underpowering [41].
jcm-12-01854-t001_Table 1Table 1Summary of the clinical study results and overview of all different interventions.EndpointTimeGroupResult*p* ValueReference**Bronchial rheoplasty**




CAT, absolute change6 months
−8.0 (−14.0 to −2.0) *0.0002[22]
6 months
−5.4 ± 7.6N/A[23]SGRQ, absolute change6 months
−7.2 (−19.8 to −3.1) *0.0002[22]
6 months
−8.0 ± 15.4N/A[23]**Metered cryospray**




CAT, absolute change3 months
−3.8 ± 7.1<0.01[24]
12 months
−2.0 ± 7.20.12[24]SGRQ, absolute change3 months
−6.4 ± 14.40.01[24]
12 months
−4.6 ± 15.10.10[24]LCQ, absolute change3 months
21.6 ± 32.2<0.01[24]
12 months
9.1 ±29.00.17[24]**Karakoca balloon desobstruction**




FEV_1_, absolute (l)baseline
0.77 ± 0.26


1 month
1.3 ± 0.5<0.001[29]Borg scale, absolutebaseline
8.7 ± 0.6


1 month
5.1 ± 0.70.002[29]6MWT, absolute (m)baseline
68.7 ± 41.4


1 month
387.4 ± 113.4<0.001[29]**Targeted lung denervation**




FEV_1_, relative change12 months20 W11.6 ± 32.3
[30]
12 months15 W0.02 ± 15.120 vs. 15 W 0.324[30]
12 months
40.3 ± 42.1<0.05[35]FEV_1_, absolute change (mL)12 months29 W57.0 ± 82.00.03[36]
12 months32 W94.2 ± 228.00.18[36]
6 monthsTLD127.6 ± 201.0Sham vs. TLD 0.345[39]
12 monthsTLD74.3 ± 213.1Sham vs. TLD 0.539[39]
2 yearsTLD−0.02 ± 0.141 vs. 2 years > 0.05[40]6MWT, absolute change12 months20 W24.2 ± 45.6
[30]
12 months15 W−9.3 ± 70.620 vs. 15 W 0.224[30]
12 months
53.7 ± 74.4<0.05[35]CAT, absolute change12 months29 W0.3 ±7.80.890[36]
12 months32 W−2.9 ± 6.10.14[36]
6 monthsTLD−2.0 ± 6.5Sham vs. TLD 0.472[39]
12 monthsTLD−0.9 ± 6.4Sham vs. TLD 0.175[39]SGRQ, absolute change12 months20 W−11.1 ± 9.1
[30]
12 months15 W−0.9 ± 8.620 vs. 15 W 0.045[30]
12 months29 W−1.9 ± 12.50.617[36]
12 months32 W−7.5 ±10.30.036[36]
6 monthsTLD−8.3 ± 12.6Sham vs. TLD 0.138[39]
12 monthsTLD−5.1 ± 14.4Sham vs. TLD 0.441[39]
2 yearsTLD1.8 ± 12.71 vs. 2 years > 0.05[40]SGRQ, relative change12 months
−1.9 ± 20.8>0.05[35]Borg scale, absolute change12 months20 W−0.3 ± 2.0
[30]
12 months15 W−0.9 ± 2.020 vs. 15 W 0.556[30]Borg scale, relative change12 months
−0.82 ± 1.83>0.05[35]mMRC, absolute change12 months20 W−0.7 ± 0.8
[30]
12 months15 W0.6 ± 1.320 vs. 15 W 0.085[30]
6 monthsTLD−0.5 ± 1.0Sham vs. TLD 0.337[39]
12 monthsTLD−0.4 ± 0.8Sham vs. TLD 0.279[39]
12 months
−0.7 ± 1.1>0.05[35]Results are presented as mean ± standard deviation, except * as median ±95% CI. CAT: COPD Assessment Test; SGRQ: St. George’s Respiratory Questionnaire; LCQ: Leicester Cough Questionnaire; FEV_1_: forced expiratory volume in 1 s; 6MWT: 6 min walking test; mMRC: modified Medical Research Council; TLD: targeted lung denervation.


A large multicenter, randomized, sham-bronchoscopy-controlled, double-blind trial enrolling 400 patients with a follow-up period of 5 years is currently recruiting (AIRFLOW-3; NCT03639051) [42].

Overall, there is already broad existing data for targeted lung denervation supporting its feasibility, acceptable safety after adapted, accurate use and comprehensive operator training, as well as long-term efficacy regarding the reduction in exacerbation events and a deceleration in lung function decline. Further information on its longer-term effects on quality of life, mortality and disease course is of clinical interest and will probably be provided by the ongoing trial.

## 3. Discussion

Currently, four different bronchoscopic treatment options for symptomatic patients with Chronic Obstructive Pulmonary Disease and features of chronic bronchitis with or without frequent exacerbations are under investigation. Targeting the phenotype of chronic bronchitis, metered cryospray and bronchial rheoplasty might be promising interventions, both concentrating on the pathophysiologic problem of increasing numbers of mucus-producing cells and improving symptoms and quality of life. Nevertheless, large randomized, controlled studies with long-term follow-up for safety and efficacy are still lacking and will be pivotal for proper patient selection and broader recommendations. For the Karakoca balloon desobstruction, only a single-center case series is available up to now, and more information must be gained for assessing the safety and efficacy of this intervention. Future studies will provide more information for comparing and weighing these different interventions and defining the most effective and safest option. More data exists for targeted lung denervation, in contrast, showing an acceptable safety profile and promising beneficial clinical outcomes, such as a reduction in exacerbations, stabilization of lung function and quality of life over time.

Overall, compliance and adherence to drug and inhaler therapy in COPD are described as low. In this context, different factors, such as demographic characteristics, socioeconomic status, satisfaction with the inhaler type and/or treatment-related side effects are pivotal [43,44,45]. Effective interventional treatment options with a good safety profile and an effect on symptoms and progression of the disease would be an essential future step for therapy, especially in patients with a high burden of disease and low self-reported quality of life, as previously seen in newer cardiologic interventional treatment options [46]. 

Given the proinflammatory autocrine and paracrine effect of goblet cells [47], potential anti-inflammatory effects and applicability of goblet cell ablating interventions in inflammatory and secretory diseases other than chronic bronchitis might be of future scientific interest after definite confirmation of their safety and efficacy. Similarly, there is already existing evidence of an anti-inflammatory effect of TLD [33,48], and future long-term investigations should assess its impact on disease progression, whereby usage even in early disease stages would be of clinical interest. Newer investigations also support its use in patients with asthma [49], but further randomized, controlled studies are still outstanding. 

Considering the widely used subjective definition of chronic bronchitis, with cough and sputum production for at least three months per year for two consecutive years [3], identifying the most suitable selection of patients and interventions will be crucial when entering the commercialization phase of these minimal invasive interventions.

In conclusion, interventional treatment options in patients with Chronic Obstructive Pulmonary Disease are an extending scientific field and future questions should be posed, especially regarding consideration of the most favorable option for each different phenotype, the long-term safety, efficacy and durability of the procedures, the possibilities for retreatment and the spectrum of other potentially treatable diseases. 

## Figures and Tables

**Figure 1 jcm-12-01854-f001:**
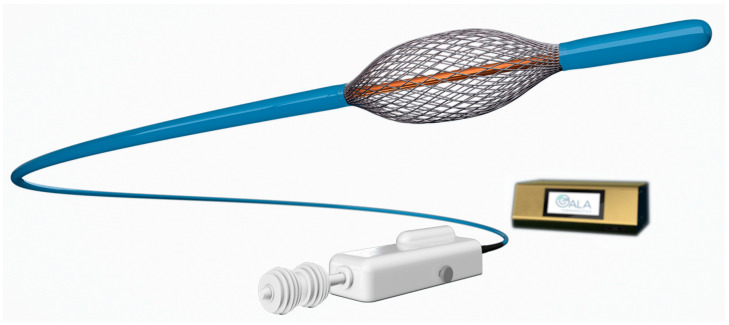
The RheOx bronchial rheoplasty catheter. Reproduced with permission from GALA Therapeutics, San Carlos, CA, USA.

**Figure 2 jcm-12-01854-f002:**
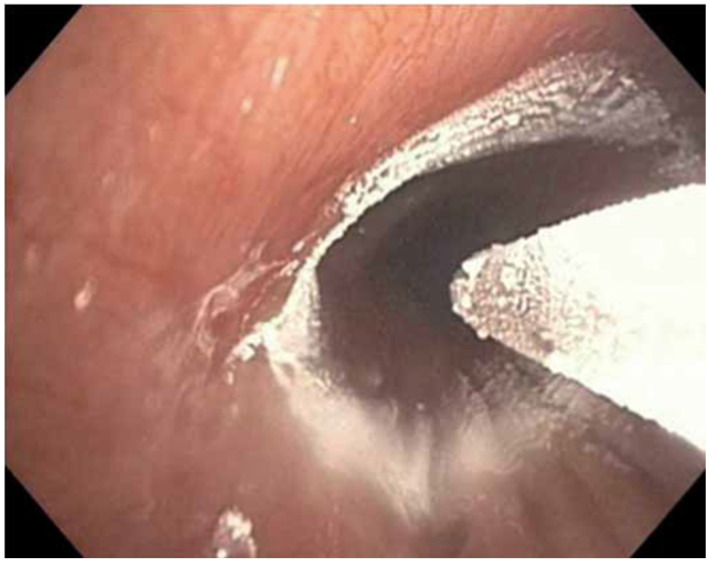
The RejuvenAir System metered cryospray catheter. Reproduced with permission of the © ERS 2023: European Respiratory Review 30 (159) 200281; DOI: 10.1183/16000617.0281-2020 Published 19 January 2021 [27].

**Figure 3 jcm-12-01854-f003:**
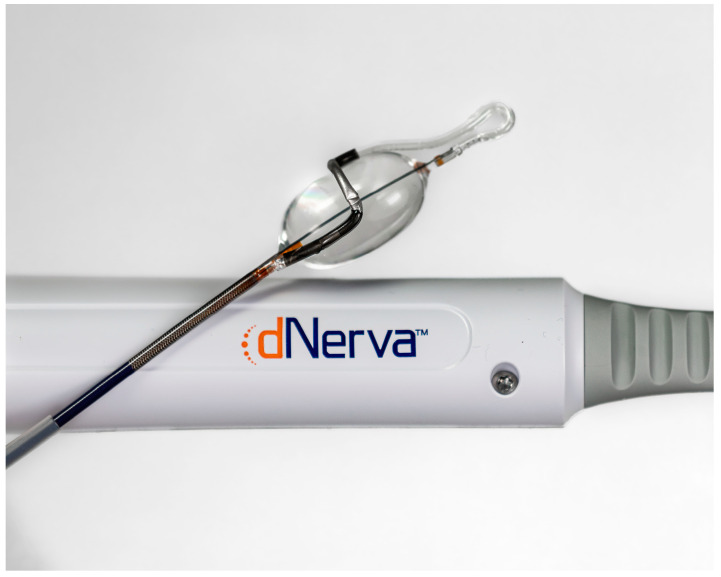
The dual-cooled radiofrequency catheter for targeted lung denervation. Reproduced with permission from Nuvaira Inc., Maple Grove, MN, USA.

## Data Availability

No new data were created or analyzed in this study. Data sharing is not applicable to this article.

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
