# Peer review of "Modern Bronchoscopic Treatment Options for Patients with Chronic Bronchitis"

_jcm, 2023, doi:10.3390/jcm12051854_

Round 1

Reviewer 1 Report

This is a review paper that address the issue of chronic bronchitis focalizing on innovative methods of treatment related to broncoscopy. Overall it is is well-organized ad well written but some concepts need to be refined.

1)    While the authors accurately describe “what” has been reported in the literature, the significance of these findings could be further emphasized.  Some sentences could be added  at the end or through each of the 4 paragraphs 2.1-2.4 emphasizing novel and/or interesting points and insights.  This could also increase the reader’s attention while reading these paragraphs.

2)    Page 1 lines 36-40 refer to a classification of COPD in 4 different phenotypes that, as described, appear unclear and poorly related to the main issue of the review. If distinction of COPD in these different phenotypes is considered important for the bronchoscopic treatment, its relevance should be clearly explained. Moreover this classification refers to old Spanish COPD guidelines (national, instead of international) that have been recently updated (Miravitlles M et al. Arch Bronconeumol. 2022 Jan;58(1):69-81).  Several prestigious studies support the important clinical role of chronic bronchitis in patients with and without COPD including: Woodruff, P. G. et al. N Engl J Med 2016: 374, 1811–1821; Allinson, J. P. et al. Am J Respir Crit Care Med 2016: 193, 662–672; de Oca, M. M. et al. The chronic bronchitis phenotype in subjects with and without COPD: the PLATINO study. European Respiratory Journal 2012: 40, 28–36.

Author Response

We agree with the reviewer's point 1 and added summarizing paragraphs at the end of each intervention section for better emphasis of the beneficial interventional effects, short conclusion and therefore better understanding.

We thank the reviewer for pointing out the unclear relation of the classification into the different phenotypes to the main issue of the review. For better clarification we adapted this section in the introduction, as the meaning of a phenotype-based therapeutic approach is of a specific importance for us. Furthermore, we added some suggested literature to further point out the impact of chronic bronchitis on the disease course.

Reviewer 2 Report

The current review focuses on bronchoscopic treatment for chronic bronchitis. The authors presented and discussed the main four techniques used in this field. Overall, the article is well-written, with good quality English and a good structure. However, the reader is left with some doubts the authors might have helped clarify further. 

Introduction: this section provides a comprehensive overview on COPD. However, it could be useful to add some data about COPD-related healthcare costs and its effects on work and productivity. 

Section 2 (Techniques): every technique is discussed in depth, with high quality evidence supporting the authors points. All the main findings of the different studies have been reported in a table. However, each subsection lacks a summarizing paragraph which might help the reader understand better the potentialities and the applicability of each method. 

Discussion: the authors assessed several interesting aspects about COPD therapy and correctly stated that further long term studies are needed in order to have a more clear picture about efficacy and safety of bronchoscopic therapies for COPD. 

In the end, the reader is left a little bit unsatisfied because, albeit a good presentation of evidence, there is no summary of them and their clinical implications. 

Author Response

In the introduction, we added some literature regarding COPD's negative impact on health-care costs.

We agree with the reviewers second point and thank for the comment, as it is of particular importance for us to provide the reader a comprehensive summary for better understanding the interventional methods in COPD. Therefore, we added a summarizing paragraph with the main points and clinical implications at the end of each intervention section.

Reviewer 3 Report

The manuscript was well written with acceptable language and layout. This is a review article and the references sought were good. 

The article can have minor revision.

(1) In 'Introduction', line 53,  ...furthermore increasing after severe events..... What is increasing ?

(2) Some typos should further be edited, e.g. in line 72, one 'moth' between...

Author Response

We thank the reviewer for pointing out some grammar and typing errors.

Regarding point 1, we deleted the end of the sentence, as it was confusing and redundant.

All typing errors found were corrected.